# Association between the Classification of the Genus of Batrachospermaceae (Rhodophyta) and the Environmental Factors Based on Machine Learning

**DOI:** 10.3390/plants11243485

**Published:** 2022-12-13

**Authors:** Qiqin Yang, Fangru Nan, Xudong Liu, Qi Liu, Junping Lv, Jia Feng, Fei Wang, Shulian Xie

**Affiliations:** 1Shanxi Key Laboratory for Research and Development of Regional Plants, School of Life Science, Shanxi University, Taiyuan 030006, China; 2School of Physical Education, Shanxi University, Taiyuan 030006, China

**Keywords:** Batrachospermaceae, environmental factors, machine learning, random forest, XGBoost

## Abstract

Batrachospermaceae is the largest family of freshwater red algae, widely distributed around the world, and plays an important role in maintaining the balance of spring and creek ecosystems. The deterioration of the current global ecological environment has also destroyed the habitat of Batrachospermaceae. The research on the environmental factors of Batrachospermaceae and the accurate classification of the genus is necessary for the protection, restoration, excavation, and utilization of Batrachospermaceae resources. In this paper, the database of geographical distribution and environmental factors of Batrachospermaceae was sorted out, and the relationship between the classification of genus and environmental factors in Batrachospermaceae was analyzed based on two machine learning methods, random forest and XGBoost. The result shows: (1) The models constructed by the two machine learning methods can effectively distinguish the genus of Batrachospermaceae based on environmental factors; (2) The overall AUC score of the random forest model for the classification and prediction of the genus of Batrachospermaceae reached 90.41%, and the overall AUC score of the taxonomic prediction of each genus of Batrachospermaceae reached 85.85%; (3) Combining the two methods, it is believed that the environmental factors that affect the distinction of the genus of Batrachospermaceae are mainly altitude, average relative humidity, average temperature, and minimum temperature, among which altitude has the greatest influence. The results can further clarify the taxonomy of the genus in Batrachospermaceae and enrich the research on the differences in environmental factors of Batrachospermaceae.

## 1. Introduction

Batrachospermaceae was established by Agardh in 1824 [1] and belongs to Rhodophyta, Florideophycidae, Batrachospermales. It is the largest family of freshwater red algae and there are currently 206 formally recognized species [2]. The members of Batrachospermaceae live in low temperature, low light, clean, high dissolved oxygen, and flowing spring or stream water [3]. Red algae (Rhodophyta) are an important part of the algal flora in the inland aquatic ecosystems, and the species diversity of freshwater red algae is mainly concentrated in Batrachospermaceae [4]. With the deterioration of the global ecological environment, the survival of Batrachospermaceae is under considerable threat. Some countries have listed its members as rare and protected species [5,6]. The studies of environmental factors and accurate classification of Batrachospermaceae are the basis for the protection and utilization of the family.

There have been many types of research about the influence of environmental factors on the geographical distribution, growth, and development of freshwater algae. Sheath and Burkholder [7] pointed out that the rapid fluctuation of physical and chemical conditions in five streams in Rhode Island is the main force determining the distribution and abundance of macroalgal species, community composition, and seasonal dynamics. Biggs and Price [8] and Biggs [9] stated that there is a strong link between the specific conductance and the surrounding algal biomes in New Zealand streams. Branco et al. [10] and Krupek and Branco [11] found that local conditions affect the spatial distribution of algal communities, such as irradiance, water velocity, and light intensity, in the central and western regions of Paraná state in southern Brazil. Jimenez and Fatjo [12] documented that *Batrachospermum gelatinosum* and *Sirodotia suecica* were clearly distinguished from *B. helminthosum* based on nutrient content in tropical high-altitude rivers in central Mexico. Carmona et al. [13] showed in their literature that the gametophytes of *S. suecica* grew in eutrophic circumstances and particular microhabitat conditions: high current rate speed, low irradiance, and shallow depth, in a high-altitude stream in central Mexico, and some morphological and reproductive characteristics seem to be adaptations to high current velocity: abundant secondary branches, spermatangia, and carpogonia. Xie found that the growth, occurrence frequency, and average cover of *B. gelatinosum* and *Sheathia arcuata* showed obvious seasonal variation in two springs from North China [14,15]. These studies show that there is a complex relationship between the growth and distribution of freshwater algae, including Batrachospermaceae, and environmental factors. Therefore, it could be speculated that environmental factors may have an impact on the classification of Batrachospermaceae.

At present, most of the classifications at the genus level of Batrachospermaceae are based on traditional morphological [16,17,18] and molecular biological methods [19,20,21]. It has not been documented to study the effect of differences in environmental factors on the classification of the genus of Batrachospermaceae based on the machine learning method.

Machine learning methods (ML) are statistical techniques originating in the field of artificial intelligence that focus on identifying complex structures, and nonlinear data, and generating accurate predictive models [22]. Compared with traditional statistical analysis methods, ML can model complex nonlinear relationships in data without meeting the restrictive assumptions required by traditional parameterization methods. ML provides techniques for dealing with high dimensions and missing, and can mine the information hidden in large databases. The rapid development of machine learning methods has been widely used in biological molecules, ecological informatics, medicine, and other fields [22,23,24].

Based on the advantages of machine learning methods, random forest (RF) and extreme gradient boosting (XGBoost) adopted in this study realized the classification of genera and the ranking of the importance of environmental factors based on the data of environmental factors. The effects of environmental factors on the growth and distribution of Batrachospermaceae were also discussed. This study will enrich the research on the taxonomy of genera of Batrachospermaceae and the influence of environmental factors on the growth and distribution of the Batrachospermaceae plants. The results of this study prove the feasibility of classifying the genera of the Batrachospermaceae plants by environmental, and may provide some reference for the conservation, restoration, and utilization of the plant resources of the Batrachospermaceae plants.

## 2. Results

### 2.1. Descriptive Statistical Analysis

Based on the latitude, longitude, and environmental factors of Batrachospermaceae, we have made a global distribution map (Figure 1) and the boxplot of environmental factors of Batrachospermaceae plants in different continents (Figure 2). It could be found that the members of Batrachospermaceae are distributed all over the world, and their distribution is of very obvious temperate nature. In addition, there are obvious differences in the distribution conditions of environmental factors of Batrachospermaceae on five continents, including altitude, average temperature, minimum temperature, and maximum temperature. The distribution range fluctuates most obviously.

There is a larger altitude distribution range of Batrachospermaceae in Asia and Oceania, but it is smaller in South America. In terms of the distribution range of average temperature, it has the largest distribution range in Asia, but the smallest in South America, and little difference in other continents. The distribution range of maximum temperature is larger in Asia, slightly smaller in Europe and Oceania, and the smallest in South America. The distribution range of average wind speed is larger in Asia and North America, while it is slightly small in Europe, and smaller in Oceania and South America.

### 2.2. Results of UMAP

The UMAP clustering results of the data of Batrachospermaceae samples between continents (Figure 3a) show that samples from different continents can be clustered, among which Oceania has the best clustering effect, and European and North American samples close in projected space. The UMAP clustering results among the genera of Batrachospermaceae (Figure 3b) show that the genera *Nothocladus* and *Virescentia* have the best clustering effect, followed by the genus *Sheathia*. The clustering effect of the genera *Batrachospermum*, *Kumanoa*, and *Sirodotia* are slightly inferior.

### 2.3. Results of Machine Learning Methods

#### 2.3.1. Results of the Random Forest

To realize the genus-level classification of Batrachospermaceae based on the geographical distribution and environmental factors, the classical random forest method in machine learning is used first. The overall area under the ROC (receiver operator characteristic) curve (AUC score) of the random forest model for the classification of Batrachospermaceae in this study reached 90.41%, which has a fairly good classification performance. Figure 4a shows the ROC curve of each genus in the random forest model. The full name of the ROC curve is the Receiver Operating Characteristic curve, which shows the tradeoff between specificity and sensitivity and is often used to evaluate the performance of models [25]. ROC curve abscess was 1-specificity, with the false positive rate (FPR) representing the proportion of positive predicted but negative samples to all negative samples. The ordinate of the ROC curve is Sensitivity, true positive rate (TPR), which represents the proportion of predicted positive samples in all positive samples that are positive [26]. Normally, the area range under the ROC curve is 0.5–1. When the area under the ROC curve is closer to 1, the better the performance of the model. The closer the area is to 0.5, the worse the model performance is and the closer it is to random prediction. Among them, the model has the best classification performance for the genera *Nothocladus* and *Virescentia*, followed by genera *Sheathia* and *Sirodotia*, and the classification effect for the genera *Batrachospermum* and *Kumanoa* is slightly lower. Figure 4b is the confusion matrix of the random forest model on the validation set. During the construction of the random forest model, the original data set is randomly divided into a training set and a validation set in a ratio of 3:1. Through the confusion matrix, the real situation can be observed when the model predicts each genus. For easy observation, we set the numbers on the diagonal of the matrix to zero. The predicted values were from the samples of the genus *Batrachospermum*, and a total of 7 samples were mispredicted, including 1 in the genus *Kumanoa*, 1 in the Remainder, and 5 in the genus *Sheathia*. By analogy, it can be found that the samples of the genus *Sheathia* are easily misidentified as the genus *Batrachospermum* by the model, and the samples of the genus *Batrachospermum* are also easily misidentified as the genus *Sheathia*. These two genera are easily confused by the model, which may be because the two genera in the geographical distribution and environmental factors are similar to each other. In addition, the samples of the genus *Kumanoa* are easily predicted to be the genus *Sirodotia*. Figure 4c is the ranking result of the importance of environmental factors on the classification results obtained in the operation of the random forest classification model. It can be found that the altitude in the model is the most important environmental factor affecting the classification results of Batrachospermaceae, followed by atmospheric pressure, average relative humidity, minimum temperature, and average temperature, and to a lesser extent, maximum sustainable wind speed, maximum temperature, and average wind speed.

#### 2.3.2. Results of the XGBoost

Previously, the random forest model was used to classify the Batrachospermaceae according to their geographical distribution and differences in environmental factors. In this section, the XGBoost method in machine learning was used to compare with the results of the random forest model. In this study, the overall area under the ROC curve (AUC score) of the XGBoost model for the classification of Batrachospermaceae reached 85.85%, which also has a relatively good classification performance. Figure 5a shows the ROC curve for each genus in the XGBoost model. Among them, the model has the best classification performance for *Nothocladus* and *Virescentia*, followed by *Sirodotia*, slightly worse classification performance for *Sheathia* and *Batrachospermum*, and the worst classification performance for *Kumanoa*. Figure 5b is the confusion matrix of the XGBoost model on the validation set. During the construction of the XGBoost model, the original data set is randomly divided into a training set and a validation set in a ratio of 3:1. Through the confusion matrix, we can observe the real situation when the model predicts each genus. To facilitate our observation of the results, we set the numbers on the diagonal of the matrix to zero for ease of observation. The predicted values were samples of the genus *Batrachospermum*, and a total of 12 samples were wrongly predicted, including 3 in the genus *Kumanoa*, 1 in the Remainder, 6 in the genus *Sheathia*, and 2 in the genus *Sirodotia*. By analogy, it can be found that in the XGBoost model, the samples of the genus *Sheathia* are easily misidentified as the genus *Batrachospermum* by the model, and the samples of the genus *Batrachospermum* are also easily misidentified as the genus *Sheathia*, which is similar to the results of the random forest model. In addition, the samples of the genus *Kumanoa* were also easily predicted to be the genera *Batrachospermum and Sheathia*. Figure 5c is the ranking result of the importance of environmental factors on the classification results obtained in the operation of the XGBoost classification model. It can be found that, similar to the random forest model, the most important environmental factor affecting the classification results of Batrachospermaceae is altitude, followed by average relative humidity, mean temperature, minimum temperature, and atmospheric pressure, and to a lesser extent, mean wind speed, maximum temperature, and maximum sustainable wind speed.

## 3. Discussion

According to the results of classification prediction by two machine learning methods at the genus level of Batrachospermaceae, the classification performance of the random forest model is better than that of the XGBoost model. The overall AUC score of the random forest model reaches 90.41%, and the XGBoost model reaches 85.85%. Based on the ranking of the importance of environmental factors of the two methods, the most important environmental factors affecting the classification results at the genus level of Batrachospermaceae are altitude, average relative humidity, average temperature, and minimum temperature.

As a special environmental factor, altitude is a comprehensive reflection of many related environmental factors including climate, geology, water chemistry, and so on [27]. In addition, changes in altitude are also closely related to changes in many environmental factors such as precipitation and temperature [28,29]. Altitude may increase the heterogeneity of environmental elements including climate and may increase the spatial isolation of species, so it can reduce the similarity between regions or communities [30]. The difference in altitude shows the difference in conditions such as precipitation and temperature, so altitude is an important factor affecting the distribution difference of different genera of Batrachospermaceae.

The relative humidity is an important physical quantity that characterizes air humidity, which indicates the degree of saturation of water vapor in the air [31]. The change of relative humidity is comprehensively affected by various conditions such as circulation form, cloud cover, precipitation, wind, and topographic factors [32]. The change of relative humidity in a region is mainly affected by local temperature, precipitation, and wind speed. Batrachospermaceae are widely distributed around the world, and the difference in average relative humidity will inevitably lead to the difference in the geographical distribution of the family.

There is a close relationship between water temperature and air temperature. Changes in air temperature can cause certain changes in water temperature in rivers and lakes [27,33]. As river and lake water temperatures are close to equilibrium with air temperature, air temperature is a key variable affecting water temperature in most biological systems, strongly affecting water chemistry, biochemical reactions, and biota growth/death [34]. Temperature conditions affect the latitude, elevation, watershed distribution, and seasonality of freshwater red algae, and some geographic patterns of freshwater red algae are also affected by photosynthesis on temperature [3].Therefore, the average temperature and minimum temperature can also affect the growth and distribution of Batrachospermaceae to a certain extent.

The results of the importance of environmental factors of the two machine learning methods in this study found that the impact of longitude and latitude on the taxonomy of Batrachospermaceae is extremely important, and it also shows that altitude is also very important for the classification of Batrachospermaceae, which is consistent with Branco et al. That is, space can have a strong impact on the community composition of less dispersive red algae taxa [35]. This study successfully realized the classification of the genus of Batrachospermaceae through environmental factors, which also indicated that the local ecological environment would have a certain impact on the growth and distribution of Batrachospermaceae plants. In the study of Abdelahad et al. [4], some morphological changes of plants in the family Batrachospermaceae also indicated that the local ecological environment would have a certain impact on the growth and distribution of giant algae.

Although both the random forest model and XGBoost model in this study have good classification prediction performance, limited by the size of the data, there is still room for improvement in the prediction accuracy. It is believed that with the increase in the availability of more and more comprehensive geographic distribution data and environmental factor data of Batrachospermaceae in the future, the machine learning classification model can classify Batrachospermaceae more effectively. In addition, benefiting from the advantages of machine learning methods, this study can accurately rank the importance of environmental factors that affect the genus-level classification of Batrachospermaceae, but it is difficult to observe more specific relationships between genus-level classifications and environmental factors. Later, we will also take other methods to analyze the relationship between the genus-level taxonomy of Batrachospermaceae and its environmental factors.

## 4. Materials and Methods

Figure 6 is the method block diagram of this study. The data set in this study is a standardized data set. Uniform manifold approximation and projection (UMAP) was first used to reduce the dimension of the data and preliminarily observe the clustering of each genus. After that, two machine learning methods based on different integration ideas were used to classify the genus of Batrachospermaceae: Random Forest (RF)—bagging integration idea and extreme gradient boosting (XGBoost)—boosting integration idea. Finally, the ROC curve was used to evaluate the classification performance of the model for each genus, and the actual situation of the model classification was observed through the confusion matrix, with the importance of environmental factors affecting the classification results being sorted.

### 4.1. Data Description

According to the collected specimens, relevant literature [10,12,13,37,38,39,40,41,42,43,44,45,46,47,48,49,50,51,52,53], and information from the Algaebase Database (https://www.algaebase.org/, accessed on 10 September 2022), a total of 16 genera and 101 species of Batrachospermaceae were sorted out in this study (Table 1). Figure 7 shows the proportion of data for each genus of Batrachospermaceae. Some species with the small number of samples are not enough for analytical research. Therefore, 6 genera, *Batrachospermum*, *Kumanoa*, *Sheathia*, *Sirodotia*, *Virescentia*, and *Nothocladus*, with more than 3 species are selected for analysis, and the rest of the genera are classified into one category denoted as Remainder. Table 2 shows the number of samples per genus in the database compiled in this study.

The latitude and longitude data of the Batrachospermaceae samples’ collection site are from https://www.gpsspg.com/, accessed on 12 September 2022, and the data of 8 important environmental factors are from https://www.wunderground.com/, accessed on 12 September 2022. Longitude, latitude, and environmental factors are expressed as follows: Lat: Latitude (°), Long: Longitude (°), ASL: Altitude (m), TM: Maximum temperature (°C), T: Average temperature (°C), Tm: Minimum temperature (°C), H: Average relative humidity (%), V: Average wind speed (km/h), VM: Maximum sustainable wind speed (km/h), SLP: Atmospheric pressure at sea level (kPa). Data for the average relative humidity and average wind speed are obtained by calculating the average of the daily mean values for the same acquisition day over five years.

### 4.2. Methods

#### 4.2.1. UMAP

Uniform manifold approximation and projection (UMAP) is a nonlinear dimensionality reduction technique based on manifold learning [54], which is widely used for visualization, exploratory data analysis, and clustering and classification tasks [55,56]. In addition to UMAP, there are many dimensionality reduction algorithms, such as the principal component analysis (PCA), multidimensional scaling, Sammon’s mapping, and T-distributed random neighbor embedding (t-SNE), etc. The performance of the UMAP algorithm significantly outranks other non-linear dimensionality reduction methods [56]. Compared with PCA, UMAP can precisely capture the nonlinear structure of large data sets [57]. Compared to t-SNE, UMAP is faster and has fewer parameters for tuning. In addition, UMAP has the advantage of being able to switch focus between local or global structures and provides a way to infer similarities and differences between clusters based on their proximity in the latent space [58]. In this study, we used UMAP to observe the inter-continental and inter-generic clustering of *Batrachospermum*.

#### 4.2.2. Random Forest

Random forest [59] is an efficient algorithm invented by Leo Breiman in 2001 based on classification and regression trees (CART) to form decision trees. It integrates the idea of random subspace [60] and bagging [61], and belongs to a supervised machine learning algorithm. The advantages of this algorithm are simple to use, with high accuracy, fewer parameters that need to be adjusted, and high operation and efficiency, therefore it can deal with high-dimensional (with many characteristic variables) data, fast training speed, and no overfitting phenomenon [62]. Due to the advantages, the random forest algorithm has been widely used in many fields [63], including ecology, bioinformatics, chemical informatics, etc. In this study, the construction and drawing of a random forest model are implemented by R language.

Construction of random forest:(1)Use the bootstrap method to resample the original data sample set X, and randomly generate K training sample sets X_1_, X_2_, …, X_K_;(2)Use each generated training set, generate the corresponding decision tree T_1_, T_2_, …, T_k_, and select “mtry” attributes (split attributes randomly selected from M attribute sets) on each intermediate node (non-leaf node). The attribute of the best splitting method in the set is used as the splitting attribute of the current node to split on this node;(3)Each decision tree grows completely without pruning;(4)Test and classify each decision tree on the original data sample set X;(5)By voting, the category with the most output from the K decision trees is taken as the category to which the original data sample set X belongs.

In building a random forest model, the two most important parameters are the number of decision trees (ntree) and the number of attributes in the split attribute set (mtry). In addition, it is necessary to set a minimum number of samples required for internal node repartition (min_n) in each leaf node. After optimizing the parameters, the parameters used in the construction of the random forest model in this study are set to mtry = 7, trees = 1000, and min_n = 6.

#### 4.2.3. XGBoost

XGBoost [64] algorithm is a tree model structure, which is an improvement of the gradient boosting decision tree (GBDT) calculation method and adopts additive learning model for optimization. The basic idea of the XGBoost algorithm is to continuously add new trees generated by feature splitting, and learn a new function for each new tree to fit the residuals of the previous round of predictions [65]. After the training, each tree will contain a leaf node according to the characteristics of the sample, and each leaf node corresponds to a score [64]. Finally, the scores corresponding to each tree are added to obtain the predicted value of the sample. The XGBoost model has the advantages of strong generalization ability, high expandability, and fast computing speed [66].

In the XGBoost algorithm, an additive model is constructed by iteration after iteration. At each iteration, a sub-prediction model is generated to correct the prediction residual of the current model for classification events, and finally, a model consisting of multiple sub-models is constructed. As shown in Equation (1):(1)$$\hat{y}=\sum_{k=1}^{K}f_{k}\left(x_{i}\right),f_{k}\in F$$
where ŷ as input samples x_i_ predictive value; f is the function space composed of all sub-prediction models f_k_. Therefore, the objective function is defined as:(2)$$\mathrm{Obj}\;=\sum_{i=1}^{n}l\left(y_{i},\hat{y_{i}}\right)\;$$
where $l\left(y_{i},\hat{y_{i}}\;\right)$ is the loss function, which measures the amount of error between the predicted value and the actual value; Ω(f) is the regular term, indicating the complexity of the sub-prediction model generated in each iteration. The complexity of each tree is defined as the following:(3)$$\Omega \left(f\right)=\gamma T+\frac{1}{2}\lambda \sum_{j=1}^{T}\omega_{j}^{2}$$

The objective function in XGBoost is defined as:(4)$$\mathrm{Obj}=\sum_{j=1}^{T}\left[G_{j}\omega_{j}+\frac{1}{2}\left(H_{j}+\lambda \right)\omega_{j}^{2}\right]+\gamma T$$

The model training process is as follows:(1)The model starts the initial iteration, and a sub-prediction model is constructed in each iteration.(2)Before each iteration, calculate the first-order and second-order gradients of the loss function under each training sample value.(3)In each iteration, with the goal of minimizing Equation (4), a decision tree is generated as a sub-prediction model, and the corresponding prediction value ω of each leaf node of the decision tree is calculated.(4)After each iteration, the newly generated model is added to the previous model. After several iterations, the final prediction model can be obtained.

In this study, the construction of the XGBoost model was implemented by R language. In establishing the XGBoost model, many parameters need to be adjusted (Table 3). The optimized parameters are set in this study, as follows: mtry = 2, trees = 1000, min_n = 4, tree_depth = 11, learn_rate = 0.00046294621410231, loss_reduction = 0.0126584260009419, sample_size = 0.790578703079373.

## 5. Conclusions

Two different machine learning methods (random forest and XGBoost) can be used to classify the genus of Batrachospermaceae plants through environmental factor data and had a good classification effect. Among them, the overall AUC score of the random forest model reached 90.41%, and the overall AUC score of the XGBoost model reached 85.85%. Thanks to the advantages of machine learning methods, the ranking of the importance of environmental factors can also be obtained in the model components. Altitude, average relative humidity, average temperature, and minimum temperature are the main factors affecting the classification results of the two models, among which altitude has the most significant influence. If more sample data and more environmental factor data can be obtained in the future, then the results of the model may have finer and more accurate results.

## Figures and Tables

**Figure 1 plants-11-03485-f001:**
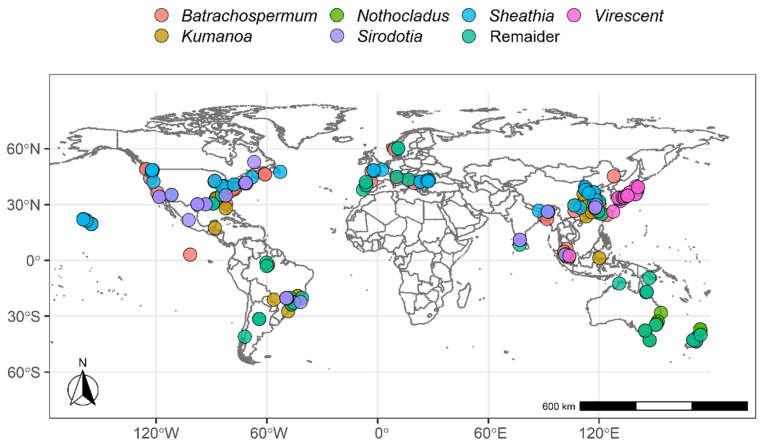
The global distribution of Batrachospermaceae.

**Figure 2 plants-11-03485-f002:**
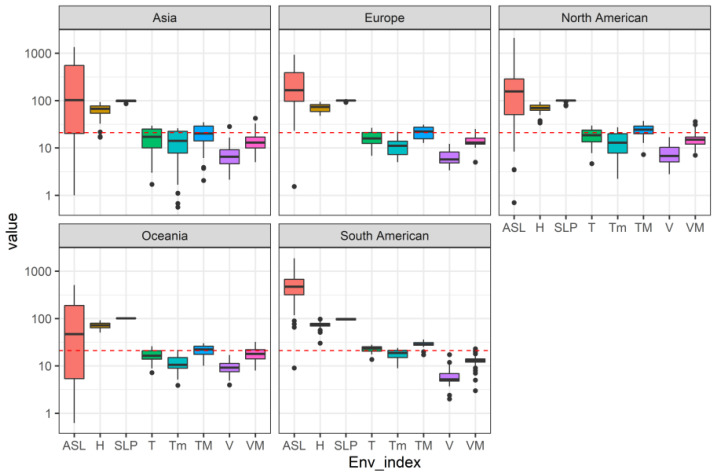
Boxplot of environmental factors of Batrachospermaceae plants in different continents. ASL: Altitude; TM: Maximum temperature; T: Average temperature; Tm: Minimum temperature; H: Average relative humidity; V: Average wind speed; VM: Maximum sustainable wind speed; SLP: Atmospheric pressure at sea level.

**Figure 3 plants-11-03485-f003:**
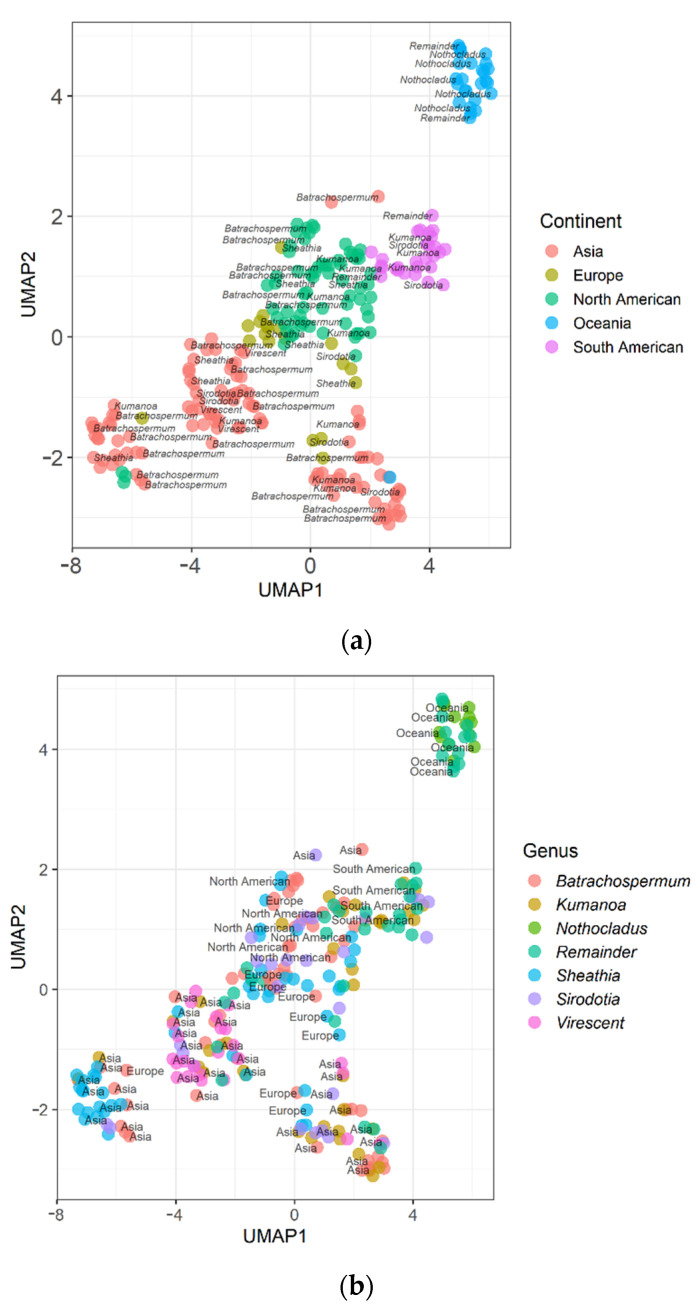
Clustering results of UMAP. (**a**) Clustering results of Batrachospermaceae among continents; (**b**) Clustering results of UMAP among the genus of Batrachospermaceae.

**Figure 4 plants-11-03485-f004:**
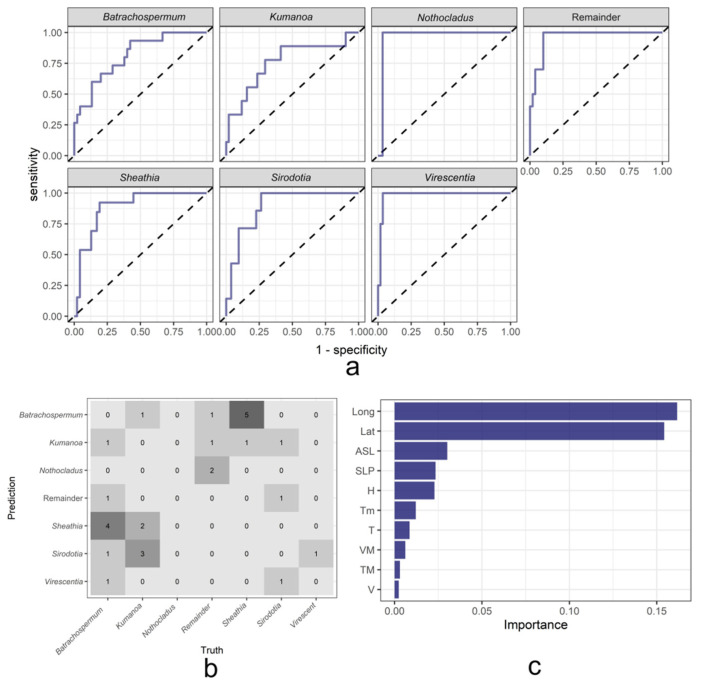
Result of random forest model. (**a**) ROC curve of each genus of random forest model; (**b**) Confusion matrix for random forest model on the validation set; (**c**) Ranking of environmental factor importance for random forest model.

**Figure 5 plants-11-03485-f005:**
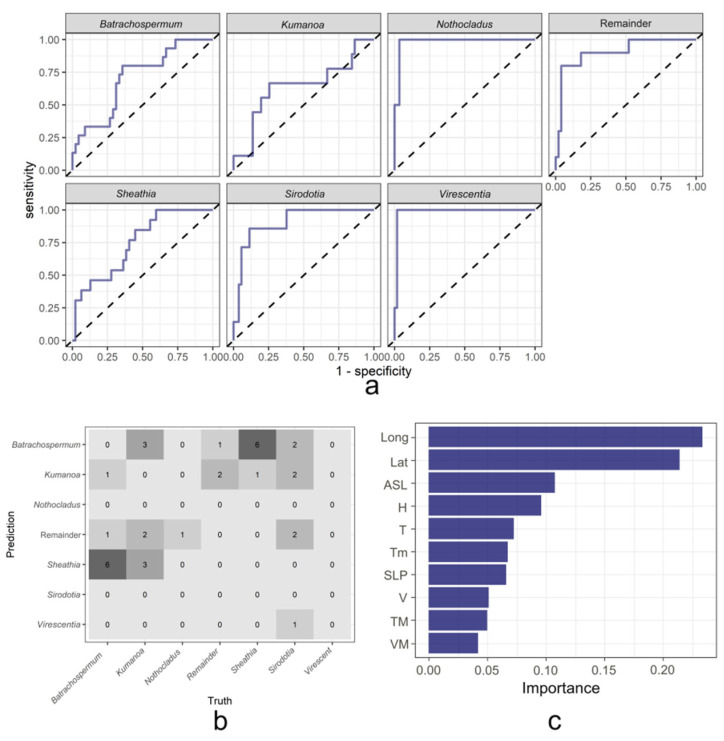
Results of the XGBoost model. (**a**) ROC curve of each genus of the XGBoost model; (**b**) Confusion matrix for XGBoost model on the validation set; (**c**) Ranking of environmental factor importance for XGBoost model.

**Figure 6 plants-11-03485-f006:**
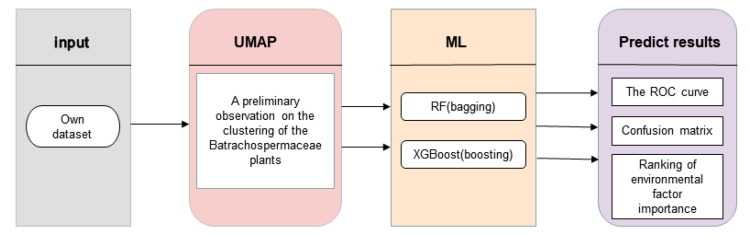
Method block diagram of this study [36].

**Figure 7 plants-11-03485-f007:**
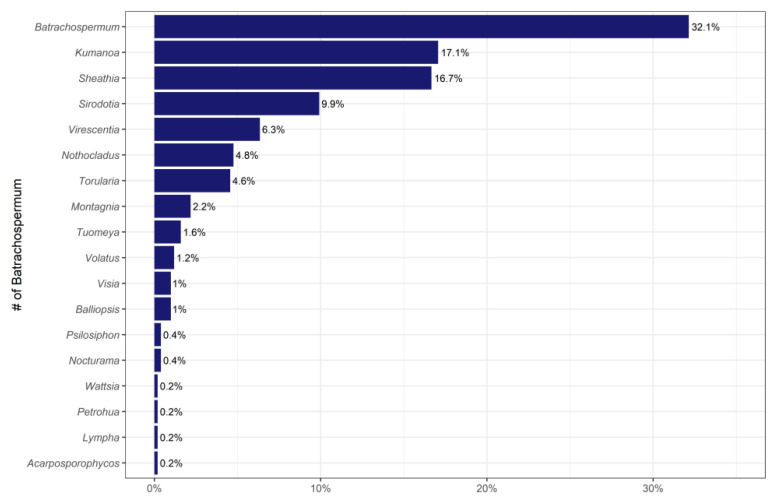
The proportion of sample data of each genus of Batrachospermaceae.

**Table 1 plants-11-03485-t001:** The number of genera and species in the database compiled in this study.

Genus	Number of Species	Genus	Number of Species
*Acarposporophycos*	1	*Psilosiphon*	1
*Balliopsis*	2	*Sheathia*	12
*Batrachospermum*	24	*Sirodotia*	9
*Kumanoa*	25	*Torularia*	3
*Lympha*	1	*Tuomeya*	1
*Montagnia*	1	*Virescentia*	4
*Nothocladus*	12	*Visia*	1
*Petrohua*	1	*Volatus*	3

**Table 2 plants-11-03485-t002:** The number of samples per genus in the database compiled in this study.

Genus	Number of Samples
*Batrachospermum*	61
*Kumanoa*	38
*Sheathia*	53
*Sirodotia*	30
*Virescentia*	24
*Nothocladus*	13
Remainder	45
Total	264

**Table 3 plants-11-03485-t003:** Main parameters.

Parameters	Illustration
mtry	The number of features to randomly take when building each tree
trees	The number of decision trees
min_n	Minimum number of samples required for internal node re-splitting
tree_depth	Maximum depth of decision tree
learn_rate	The learning rate in the ensemble, which is also the weight reduction factor for each weak classifier
loss_reduction	Minimum loss reduction required to make a further partition on a leaf node of the tree
sample_size	Subsample ratio of the training instance

## Data Availability

Not applicable.

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
