# Peer review of "Association between the Classification of the Genus of Batrachospermaceae (Rhodophyta) and the Environmental Factors Based on Machine Learning"

_plants, 2022, doi:10.3390/plants11243485_

Round 1
Reviewer 1 Report
The manuscript entitled “Association between the classification of the genus of Batrachospermaceae (Rhodophyta) and the environmental factors based on machine learning” presents well-formalized research. The structure of the article is clear, the methodology is also good, and the experimental data is also elaborately explained, but still there are a few points that needs to be addressed. I will recommend minor revisions with a few major steps that need to be evaluated for the better contribution and novelty claimed by the authors. If the authors successfully incorporate the below points, the manuscript can be considered as suitable for publication.
Authors must state their contribution in bullets points in the last of introduction section.
There are many spelling mistakes and grammatical error, i.e. illustration spelling in Table 2. etc.
It would be better if author make a block diagram of whole methodology. for reference please check. cite and follow the article. https://doi.org/10.3390/electronics10233026
Most of the references are old, very few articles cited from the last three years literature. add few more citations.
It would be better to write a pseudocode instead of the training points starting from line 309.
For the Data Availability Statement: if the data is openly available, give the proper link etc. for the readers to understand and implement further methodologies on the data.
I suggested authors to compare their results in a table with others existing studies on the identical dataset.
Author Response
Please look for the attached file.

Reviewer 2 Report
This study tried to clarify the relationship between the classification of batrachospermacean genera and environmental conditions of their habitats based on two machine learning methods. It is known that these freshwater red algae are distributed in various habitats and that some species require specific environmental conditions to grow. It is an interesting trial to apply machine learning method for macroalgal ecology and this study involves a uniqueness and novelty in this aspect. However, the explanation of methods and results are insufficient and some of the citations are questionable. The authors do not fully evaluate the present results by comparing with previous studies. Therefore, major revision should be required before publication of this study. I am not familiar with these analytical methods and have few comments on the validity of methodology. My detailed comments are listed below.
L33; According to AlgaeBase 206 species are formally accepted in Batrachospermaceae.
L33-34; “The members of Batrachospermaceae live ... stream water.” The reference should be cited.
L35-37; But Lyons et al. (2014) does not mention the role of freshwater algae at all.
L64; There are many recent molecular phylogenetic papers the authors should cite.
L73-74; But Sun et al. (2020) introduce the application of ML only to plant molecular studies.
L116-122; Many readers must be unfamiliar with this analytical method, so it is helpful to explain how to interpret the results of ROC curve, including the mean of specificity and sensitivity.
L121; Virescent should be replaced by Virescentia throughout the text and figures.
L122-133; I cannot understand this confusion matrix well. For example, a total of seven Batrachospermum samples were mispredicted, so how many Batrachospermum samples were truly predicted?
L206-209; Generally temperature of spring water is not affected by air temperature and constant throughout the year.
L224-226; How many samples were analyzed in each species? It is known that environmental condition varied even within the same species, so it is essential to refer the information of habitats for each species as much as possible. No detailed information about distribution is provided in AlgaeBase, and the authors should refer twenty or more literatures, not only three, for the present analyses.
L227-228; If so, how many samples are enough for the analytical research?
L233-237; Was the average value obtained by averaging one-year or several-years data? It is necessary to explain how to calculate average relative humidity and average wind speed, both of which frequently change with a day.
L326-337; This is not a conclusion but just a summary of the results. The authors should compare the present results with previous works based on statistical analyses (e.g., Branco et al. 2014. Hydrobiologia 732:123-132; Abdelahad et al. 2015. Eur J Phycol 50:318-329) and discuss the distinction between them. Although both models had the best classification performance for Nothocladus and Virescentia, I cannot figure out which environmental factors are useful for classifying the two genera.
Figure 2 "Box plot of the distribution of Batrachospermaceae in each continent" The sentence is not fit for this graph. Each abbreviation should be also explained in the legend.
Author Response
Please look for the attached file.
